# Atherosclerosis Development and Progression: The Role of Atherogenic Small, Dense LDL

**DOI:** 10.3390/medicina58020299

**Published:** 2022-02-16

**Authors:** Jelena Vekic, Aleksandra Zeljkovic, Arrigo F. G. Cicero, Andrej Janez, Anca Pantea Stoian, Alper Sonmez, Manfredi Rizzo

**Affiliations:** 1Department of Medical Biochemistry, University of Belgrade-Faculty of Pharmacy, 11000 Belgrade, Serbia; jelena.vekic@pharmacy.bg.ac.rs (J.V.); aleksandra.zeljkovic@pharmacy.bg.ac.rs (A.Z.); 2Department of Medicine and Surgery Sciences, IRCCS Azienda Ospedaliero, University of Bologna, 40126 Bologna, Italy; arrigo.cicero@unibo.it; 3Department of Endocrinology, Diabetes and Metabolic Diseases, University Medical Centre Ljubljana, University of Ljubljana, 1000 Ljubljana, Slovenia; andrej.janez@kclj.si; 4Faculty of Medicine, Diabetes, Nutrition and Metabolic Diseases, Carol Davila University, 050474 Bucharest, Romania; ancastoian@yahoo.com; 5Department of Endocrinology and Metabolism, Gulhane School of Medicine, University of Health Sciences, Ankara 06018, Turkey; alpersonmez@yahoo.com; 6Department of Health Promotion, Mother and Child Care, Internal Medicine and Medical Specialties, University of Palermo, 90100 Palermo, Italy

**Keywords:** small dense LDL, lipoproteins, CVD, residual risk, atherosclerosis, prevention

## Abstract

Atherosclerosis is responsible for large cardiovascular mortality in many countries globally. It has been shown over the last decades that the reduction of atherosclerotic progression is a critical factor for preventing future cardiovascular events. Low-density lipoproteins (LDL) have been successfully targeted, and their reduction is one of the key preventing measures in patients with atherosclerotic disease. LDL particles are pivotal for the formation and progression of atherosclerotic plaques; yet, they are quite heterogeneous, and smaller, denser LDL species are the most atherogenic. These particles have greater arterial entry and retention, higher susceptibility to oxidation, as well as reduced affinity for the LDL receptor. Increased proportion of small, dense LDL particles is an integral part of the atherogenic lipoprotein phenotype, the most common form of dyslipidemia associated with insulin resistance. Recent data suggest that both genetic and epigenetic factors might induce expression of this specific lipid pattern. In addition, a typical finding of increased small, dense LDL particles was confirmed in different categories of patients with elevated cardiovascular risk. Small, dense LDL is an independent risk factor for cardiovascular diseases, which emphasizes the clinical importance of both the quality and the quantity of LDL. An effective management of atherosclerotic disease should take into account the presence of small, dense LDL in order to prevent cardiovascular complications.

## 1. Introduction

The disorders of lipoprotein metabolism are considered as important factors for the initiation and progression of atherosclerosis, and analysis of standard lipid status parameters is a mandatory step in cardiovascular disease (CVD) risk assessment, prevention and therapeutic management. Based on the firm evidence that low-density lipoproteins (LDL) play the principal role in atherogenesis [1], LDL-cholesterol (LDL-C) level reduction represents the main therapeutic target for primary and secondary prevention of CVD in clinical practice guidelines [2,3]. Nevertheless, evaluation of standard lipid profile does not provide an insight into the qualitative characteristics of lipoprotein particles, nor indicate specific disorders of lipoprotein metabolism. Namely, plasma LDL pool is composed of heterogeneous mixture of particles differing in structure, density, size and atherogenic properties [4]. Among them, small, dense LDL (sdLDL) seems to be the most atherogenic fraction [5].

Plasma preponderance of sdLDL particles is strongly associated with metabolic perturbations characteristic for insulin resistance, while their causative role in atherogenesis mainly relies on rapid accumulation within arterial wall and enhanced oxidation [6]. An increased proportion of sdLDL could indicate a hidden cardiovascular risk in otherwise asymptomatic subjects [7]. Furthermore, a residual cardiovascular risk, attributable to sdLDL, still might persist even if the patient achieves its LDL-C target value [8]. These data suggests that a finding of increased sdLDL could guide the treatment individualization in specific groups of patients with metabolic disorders. Although the roles of sdLDL in the development and progression of atherosclerosis have been elucidated, clinical significance of their assessment is still controversial. In this review we will discuss the importance of sdLDL assessment for CVD risk prediction in various populations of patients, as well as their atherogenic properties and the main factors involved in their formation.

## 2. Formation and Progression of Atherosclerotic Plaque: The Role of Small, Dense LDL

Traditionally, the role of LDL in atherogenesis has been exclusively observed from the aspect of cholesterol transport and its accumulation in vascular intima, but in recent years the focus was shifted towards other mechanisms by which LDL particles promote development and progression of atherosclerosis. According to the current concept, atherosclerosis represents a systemic, chronic, inflammatory disease of arterial wall [9]. Essentially, atherosclerosis represents an inflammatory response to endothelial injury, triggered by simultaneous interplay of multiple risk factors, among which dyslipidemia and oxidative stress (OS) play major roles [10]. Both, dyslipidemia and OS act synergistically in the maintenance of inflammation during the progression of atherosclerosis [6]. Their interaction is primarily manifested by changes in the structure and function, i.e., the quality of lipoprotein particles. Namely, alterations of lipoproteins’ structure further affect their functional characteristics, and consequently their role in atherogenesis [11].

LDL particles are produced within very low density lipoproteins (VLDL)-LDL delipidation cascade and represent the main carriers of cholesterol in plasma. Each LDL particle contains one apolipoprotein B (apoB) molecule, recognized by specific LDL receptors. In addition to receptor-dependent mechanism, LDL can be removed from the bloodstream by an alternative route. Circulating LDL particles are highly heterogeneous with respect to their density, size and lipid content, which gives a rise of several distinct subfractions [4]. The particles with diameter smaller than 25.5 nm and density higher than 1.034 kg/L are denoted as sdLDL [12]. The formation of sdLDL is greatly enhanced by elevated concentration of triglycerides (TG).

In particular, hypertriglyceridemia is characterized by increased generation and/or decreased catabolism of VLDL particles. It is also associated with enhanced activity of cholesterol-ester transfer protein (CETP), which mediates exchange of core TG within VLDL and cholesteryl-esters within LDL particles, resulting in TG-enrichment of LDLs. The latter are good substrates of hepatic lipase (HL), which further converts them into smaller and denser forms [4] (Figure 1). Small, dense LDL particles are integral part of the lipid triad or atherogenic lipoprotein phenotype (ALP), which also includes elevated TG and reduced high density lipoproteins (HDL) levels [13]. This characteristic lipid pattern is a common feature of dyslipidemia in patients with metabolic syndrome and type 2 diabetes mellitus (DM) [14,15].

## 3. Genetic and Epigenetic Factors Affecting the Formation of Small, Dense LDL

To date, many attempts have been made to reveal the underlying genetic causes of ALP expression. In this regard, several cross-sectional studies suggested associations between the single nucleotide polymorphisms (SNPs) of *CETP*, *APOA5*, *APOE*, *LPL* and *LIPC* genes with the preponderance of sdLDL and/or smaller LDL size [16]. As a result, ALP is considered as a complex, polygenic trait, which includes genes that influence variations in several lipid parameters, but also genes that impact LDL size independently of TG levels; indeed, the prevalence of small, dense LDL seems to be more frequent in some ethnicities [17]. Large-scale genome-wide association studies (GWASs) have identified several candidate genes related to variation in plasma TG levels, which may also account for the variations in LDL size, thereby providing novel opportunities for personalized risk assessment, as well as innovative therapeutic targets [18].

Taking into account the fact that ALP is a complex, multifactorial phenotype, it is reasonable to assume that it will depend on interaction of both genetic and environmental factors. Indeed, heritability of LDL particle size phenotypes ranges from 40–60% [16], suggesting additional impact of environmental factors. In recent years the emerging role of epigenetic modulation and their role in the expression of genes involved in lipid metabolism, such as DNA methylation, histone modification and non-coding RNAs, has also been increasingly recognised. Epigenome wide association studies (EWASs) demonstrated that DNA methylation levels might contribute to variations in plasma lipid levels and lipid composition of different lipoprotein subclasses. These studies repeatedly singled out associations of carnitine palmitoyltransferase-1A (*CPT1A*) and ATP binding cassette transporter G1 (*ABCG1*) genes with the components of ALP.

In particular, higher methylation of *CPT1A* gene was associated with overall decreased number of VLDL, LDL and especially sdLDLs [19], as well as with reduced TG content within sdLDL particles [20]. Similarly, higher methylation of *ABCG1* gene was associated with increased TG and decreased HDL-C level [20]. The observed associations were recently confirmed in a large multi-ethnic EWAS study of leukocyte DNA methylation and blood lipids [21]. Furthermore, numerous non-coding RNAs have been identified to regulate gene expression by targeting specific mRNAs. Among the studies investigating the role of microRNAs (miRNAs) in the regulation of lipoprotein metabolism, mir-33a and mir-33b seem to be the most convincingly associated with insulin resistance and lipid profile alterations that resemble ALP, as recently reviewed in [22]. Knowing that epigenetic processes can be reversible, future studies investigating their potential roles as targets for therapeutic modulation are warranted.

## 4. Atherogenic Properties of Small, Dense LDL and the Association with Cardiovascular Risk

Several pathophysiologic mechanisms have been proposed for the explanation of enhanced atherogenic potential of sdLDL particles (Figure 2).

First, the exposure of apoB on the surface of sdLDL particles is reduced, which accounts for their reduced affinity for LDL receptors and consequent prolonged plasma retention. On the other side, their smaller size facilitates transendothelial transport and entry into the subendothelial space. Finally, sdLDLs are more vulnerable to oxidative modifications, as a consequence of their reduced antioxidant content [16]. Bearing in mind high concentration of prooxidans in the subendothelial space, sdLDL particles can easily become oxidized. Although the most important pro-atherogenic property of oxidized LDL particles (oxLDL) is their preferential uptake by macrophages and consecutive foam cell formation, these particles have a much broader spectrum of detrimental effects during atherogenesis. Indeed, once formed, oxLDL particles possess strong immunogenic, pro-inflammatory and prothrombotic potential [23], and it has been shown that their levels are closely associated with pro-atherogenic inflammatory adipokines, such as resistin [24]. Even in the earliest stages of atherogenesis, oxLDL particles are able to activate endothelial cells and stimulate the release of chemoattractant proteins, which further attract circulating immune cells into subendothelial space [25].

Moreover, oxLDL supports differentiation of monocytes into macrophages and stimulates them to secrete pro-inflammatory mediators, produce reactive oxygen species (ROS) that contribute to OS exacerbation, as well as growth factors that stimulate cell proliferation [23]. In addition, these cells increase synthesis of extracellular matrix components and matrix-metalloproteinases, which stimulate growth of atherosclerotic lesion, but also contribute to plaque instability [26]. Overall, oxidized LDL particles can trigger a whole series of successive events, forming numerous vicious circles that lead to the progression of atherosclerosis. Hence, sdLDLs, serving as the precursors of oxLDLs, should be considered as one of the main driving forces of atherosclerosis initiation and progression.

Large prospective studies of LDL particle distribution had long ago revealed the prevalence of sdLDLs in CVD patients and confirmed their independent contribution to CVD development [27,28,29]. A recent meta-analysis pointed towards sdLDL as a biomarker of increased cardiovascular risk [30]. Additionally, it has been shown that higher prevalence of sdLDL particles independently contributes to the risk for ischemic stroke development [31,32,33]. Yet, current American College of Cardiology (ACC)/American Heart Association (AHA) and European Society of Cardiology (ESC)/European Atherosclerosis Society (EAS) clinical practice guidelines on the management of blood cholesterol and primary prevention of CVD do not recommend estimation of sdLDL particles, nor specifically label sdLDL as a therapeutic target [2,3,34,35]. On the other hand, a recent large-scale prospective study conducted in Japanese community indicates the clinical relevance of sdLDL-cholesterol measuring [36]. Thus, the question of the possible medical benefit obtained by the routine assessment of sdLDL, that we raised in 2009 [11], remains open even today.

## 5. Small, Dense LDL in Obesity, Diabetes and Chronic Kidney Disease

Apart from clearly demonstrated abundance of sdLDLs in patients with CVD, many previous studies have explored the distribution of LDL subclasses in other conditions that are associated with atherosclerosis development. Obesity and insulin resistance are among principal atherogenic risk factors, and it has been shown that the accumulation of sdLDL particles is among hallmarks of obesity-induced dyslipidemia [22]. Moreover, it has been demonstrated that the predominance of sdLDLs is the most important predictor of metabolically unhealthy overweight phenotype [37]. Similarly, we demonstrated increased prevalence of sdLDL particles in metabolically unhealthy overweight subjects than in their metabolically healthy counterparts, although similar pattern could not be confirmed among metabolically healthy and unhealthy obese subjects [38].

In contrast to numerous studies of lipoprotein subclasses distribution in overweight and obese adults, there is a lack of similar research in pediatric population. Still, it has been shown that higher prevalence of sdLDL subclasses is present in obese children [39], thus implying that adverse shift towards more atherogenic particles starts in childhood. In line with this, we reported greater abundance of sdLDL particles in hypertensive than in normotensive obese children [39]. It should also be noted that the investigations have shown elevated presence of sdLDLs in subjects with obstructive sleep apnea (OSA), which is another disorder associated with obesity and metabolic disturbances [40].

DM is considered as one of principal cardiovascular risk factors. Adverse distribution of LDL subclasses, characterized by higher presence of sdLDL species, was demonstrated in both type 1 and type 2 DM patients, as well as in gestational diabetes [14,41], and several antidiabetic therapies have been investigated over the years for their potential role in increasing LDL size [42,43]. Some nutraceuticals are also effective on the oxidation of LDL and, therefore, on their subclass distribution [44,45]. Moreover, it has been shown that sdLDL-cholesterol level is associated with type 2 DM predecessor—metabolic syndrome, independently of obesity and inflammation [46]. We have demonstrated that the relation between proprotein convertase subtilisin/kexin type 9 (PCSK9) and sdLDL particles in type 1 DM youths is modulated by the extent of achieved glycemic control [47]. Altogether, these findings might implicate that atherogenic dyslipidemia in diabetes has a more complex etiology than it could be assumed by relaying only on the presence of hyperglycemia, obesity, or inflammation. It remains to be established how such complex constellation of metabolic traits could impact CVD risk in diabetic subjects.

CVD is the most frequent complication of chronic kidney disease (CKD) and dyslipidemia is amongst major contributing CVD risk factors in renal disease patients. Increased prevalence of sdLDLs is common in CKD [48]. Importantly, it has been shown that the quantity of sdLDL particles is associated with the disease stage in both adult and young CKD patients [49,50]. In addition, sdLDL-cholesterol level was identified as an independent risk factor for CVD development in subjects with CKD [44]. Of note, our research is this field have demonstrated significant association among genotypes of gluthatione-S-transferase characterized by low anti-oxidative activity and high proportion of sdLDLs in hemodialysis patients, which might suggest even stronger increase of CVD risk [51]. Additionally, we have reported increased sdLDL-cholesterol and sdLDL-apolipoprotein B concentrations in pediatric renal transplant recipients, whilst the levels of these two parameters were dependent on the type of immunosuppressive therapy [52]. Thus, not only CKD, but also related therapeutic options might increase overall cardiovascular risk in this vulnerable group of patients.

## 6. Small, Dense LDL and Endocrine Disorders, Pregnancy and Menopause

Chronic inflammatory diseases are known for increased CVD risk, even though LDL-cholesterol level is not necessarily raised in these subjects. Yet, it has been found that sdLDL/LDL-cholesterol ratio is higher in patients with various chronic inflammatory diseases [53]. SdLDL particles were reportedly more prevalent in patients with psoriatic arthritis [54] and altered LDL subclasses distribution has been observed in rheumatoid arthritis too [55]. Similarly, smaller LDL size and increased proportion of sdLDL particles has been reported in sarcoidosis patients [56,57]. Therefore, available evidence suggests that adverse LDL subclasses distribution could be one of the key features of elevated CVD risk in patients with inflammatory diseases.

Among endocrine disorders, hypothyroidism is most frequently associated with metabolic changes and cardiovascular risk factors. The principal mechanisms that link hypothyroidism and dyslipidemia are diminished synthesis of LDL receptor and resultant reduced clearance of LDL particles [58], thus accumulation of proatherogenic lipid species is common in this condition. Indeed, increased sdLDL-cholesterol level was observed in hypothyroid patients, even if their routine lipid profile parameters were in reference range [59]. Similar results were found in patients with subclinical hypothyroidism [60], thereby suggesting the need for advanced lipid profiling to prevent CVD development in this group of patients.

Polycystic ovary syndrome (PCOS) is another endocrine disorder associated with insulin resistance, metabolic syndrome, and elevated CVD risk [61]. Previous studies have shown that LDL size is diminished and proportion of sdLDL particles is increased in women with PCOS [62,63]. Smaller LDL particle size and higher apolipoprotein A-I levels are independent of triglyceride levels. After adjusting for triglyceride levels, other traits of insulin resistance syndrome (IRS) were not associated with LDL size phenotype, suggesting that the IRS-related sdLDL is linked most strongly to alterations in triglyceride levels [64]. It is important to mention that inconsistent results were achieved when the contribution of obesity to the development of atherogenic dyslipidemia in patients with PCOS was analyzed [65,66], so further studies are needed to elucidate this topic. Of note, our recent findings [67] suggest that PCOS patients with predominant sdLDL particles have diminished antioxidative capacity conferred to small HDL subclasses, which might provide an additional explanation for the increased CVD risk in this large population.

Pregnancy affects lipid profile as well. Even though these changes are largely reversible, they might contribute to the development of pregnancy complications and to the increased cardiovascular risk in later life [68]. Metabolic adaptation in pregnancy creates an intravascular milieu that favours formation of sdLDL particles. We have shown that LDL particle size decreases gradually during pregnancy and that increased presence of sdLDLs is associated with smaller weight, length and head circumference of newborns [69]. Furthermore, it has been shown that sdLDLs are dominant LDL subclasses in cord blood, and clinical relevance of these findings remains to be fully elucidated [70].

It should be noted that recent findings of Li and colleagues [71] suggest the absence of any relation between sdLDL-cholesterol concentration and pregnancy complications, although the level of sdLDL-cholesterol was relevant for the delivery mode. Similarly, we found no consistent differences in LDL subclasses distribution among women with normotensive pregnancy and preeclampsia [72]. Yet, it is worth of mentioning that recent findings indicate a disbalance in cholesterol synthesis and absorption in women with preeclampsia, so the contribution of profound changes of LDL metabolism to the development of pregnancy complications is still to be resolved [73].

The risk for CVD is significantly increased in menopausal women, largely due to adverse changes in lipid profile. Prevalence of sdLDL subclasses was recognized as a typical feature of menopausal lipid profile [74,75,76]. A recent large-scale study conducted in Japanese population revealed a pattern of sdLDL-cholesterol and sdLDL-cholesterol/LDL-cholesterol ratio changes during a lifespan of a woman [77]. The authors demonstrated an increase in sdLDL-cholesterol and sdLDL-cholesterol/LDL-cholesterol ratio until the age of approximately 65. The level of sdLDL-cholesterol followed a downward trend in women older than 65 years. However, sdLDL-cholesterol/LDL-cholesterol ratio reached a peak value in the age of 65–69 and remained stable afterwards. Therefore, the age and gender-associated patterns of LDL subclasses distribution additionally confirms the impact of this specific lipid moiety to the modulation of CVD risk.

## 7. Conclusions

Overall, sdLDL represents a novel critical biomarker, which can help to identify individuals at higher risk for the development of CVD, who would benefit from more intensive preventive and therapeutical measures, in order to reduce their residual risk to undergo major cardiovascular events (Figure 3). This is of greater importance now due to coronavirus COVID-19 pandemic, since it has been noticed in many geographical areas a significant increase of CVD and related mortality [78]. Insights and learnings from such experience can guide us for a better management of CVD risk [79] and applying in clinical practice the best available tools for proper risk prevention, such as the measurement of sdLDL.

It is important to mention that several methods have been introduced for the analysis of sdLDL particles, but definitive test for their assessment has not been established so far. In the light of new evidence, advanced lipid testing, including sdLDL measurement should be implemented in clinical laboratories as a part of a specialized lipid profiling. Such approach would make this relevant clinical information available to clinicians, improving current preventive and therapeutic measures and, ultimately, reducing the cardiovascular burden of many patients.

## Figures and Tables

**Figure 1 medicina-58-00299-f001:**
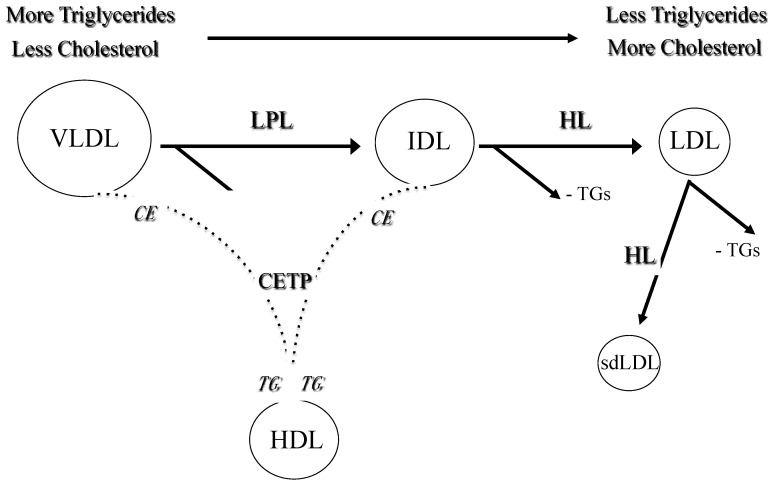
The formation of small dense LDL. Legend: VLDL: very low density lipoproteins; IDL: intermediate density lipoproteins; LDL: low density lipoproteins; HDL: high-density lipoproteins; TG: triglycerides; CE: cholesterol esters; LPL: lipoprotein lipase; HL: hepatic lipase; CETP: cholesterol-ester transfer protein.

**Figure 2 medicina-58-00299-f002:**
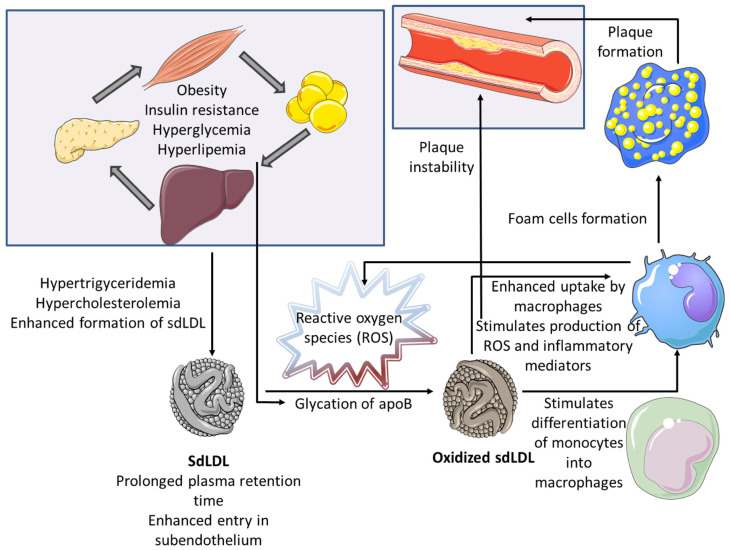
Pathophysiologic mechanisms explaining the enhanced atherogenic potential of small dense LDL.

**Figure 3 medicina-58-00299-f003:**
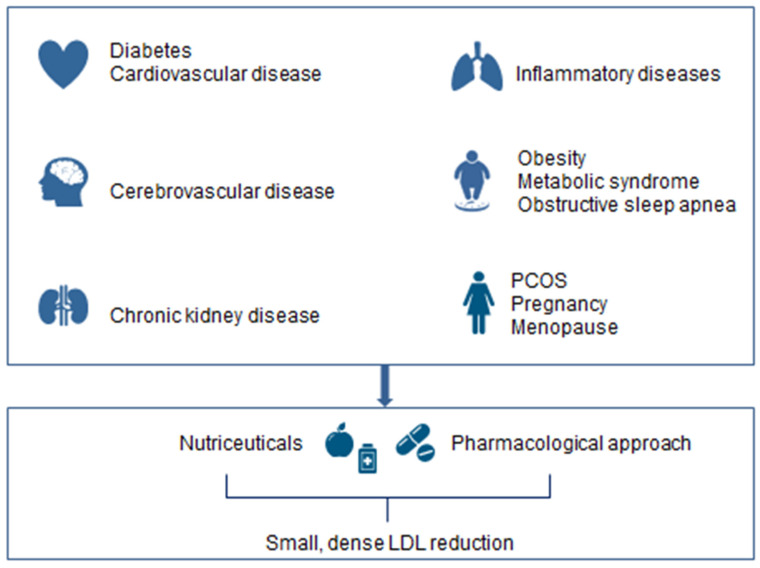
The clinical significance of small dense LDL.

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
