# Peer review of "Atherosclerosis Development and Progression: The Role of Atherogenic Small, Dense LDL"

_medicina, 2022, doi:10.3390/medicina58020299_

Round 1
Reviewer 1 Report
Vekic et al. summarized the role of the small dense LDL in the development of atherosclerosis in different comorbidities in a short review. The topic is interesting and novel, however, Figures are needed to visualize and summarize the content of this review (e.g., Fig. 1 could be structure/important molecular properties of sdLDL, Fig. 2 might contain the pathomechanistic role of sdLDL in the development of atherosclerosis and interplay with other factors such as oxidative stress, inflammation, etc., and comorbidities. In the present form, it is hard to follow the text only.
Reviewer 2 Report
-Good work but you might need to write the full name for each abbreviation, ex: VLDL, HDL-C,ACC/AHA and ESC/EAS.
- Explain the sentence more ( in TG-enrichment of LDLs. The latter are good substrates of hepatic lipase (HL), which further converts them into smaller and denser forms). Maybe drawing a scheme of that would be great.
Round 2
Reviewer 1 Report
The authors prepared the requested Figures and answered my questions.
I suggest the publication of their MS.